# A Case of Acquired Aplastic Anemia after Severe Hepatitis- Probably Induced by the Pfizer/BioNTech Vaccine: A Case Report and Review of Literature

**DOI:** 10.3390/vaccines11071228

**Published:** 2023-07-11

**Authors:** Zahra Kmira, Khirallah Sabrine, Guermazi Monia, Akkari Imen, Chiba Dorra, Bannour Rania, Fathallah Neila, Bouteraa Walid, Zaier Monia, Ben Youssef Yosra, Regaieg Haifa, Khelif Abderrahim

**Affiliations:** 1Department of Clinical Hematology, Farhat Hached University Hospital, Sousse 4000, Tunisia; sabrine.khirallah@gmail.com (K.S.); mannou.guermazi@gmail.com (G.M.); bouteraawalid@yahoo.fr (B.W.); zaiermonia@yahoo.fr (Z.M.); yosra.benyoussef69@gmail.com (B.Y.Y.); regaieghaifa@yahoo.fr (R.H.); abkhelif@yahoo.com (K.A.); 2Departement of Gastrology, Farhat Hached University Hospital, Sousse 4000, Tunisia; imenakkaribm@gmail.com; 3Department of Pathology, Farhat Hached University Hospital, Sousse 4000, Tunisia; dorrachiba.dc@gmail.com; 4Department of Hygiene, Sahloul University Hospital, Sousse 4000, Tunisia; raniaa.bannour@gmail.com; 5Department of Pharmacology, Farhat Hached University Hospital, Sousse 4000, Tunisia; neilafathallah@gmail.com

**Keywords:** aplastic anemia, acute hepatitis, Pfizer/BioNTech vaccine

## Abstract

Introduction: An important but rare adverse effect of vaccines is their association with autoimmune events, including hepatitis and aplastic anemia (AA). In this paper, we report a case of hepatitis followed by AA that occurred after the COVID-19 vaccine was administered. Case report: This paper focuses on a 30-year-old female who presented with acute hepatitis three weeks after receiving the second dose of the coronavirus Pfizer/BioNTech vaccine. After an extensive diagnostic evaluation was conducted that did not discover a specific cause, the Pfizer/BioNTech vaccine was suspected and the patient was treated with corticosteroids. One week after the onset of a liver disorder, the patient presented with gum bleeding and pancytopenia, and the diagnosis of AA was established via laboratory testing and bone marrow biopsy. After the diagnosis, the patient received immunosuppressive therapy using anti-lymphocyte serum (ATGAM) and CYCLOSPORINE A with progressive improvements in cytopenia. The important issue is whether AA is related to acute hepatitis or the coronavirus vaccine. Conclusion: Clinicians should be aware of the risk of both the possibility of acute hepatitis, AA, or both after receiving the COVID-19 vaccination. It is very hard to distinguish the cause of AA between vaccine- and hepatitis-related AA. Predicting who develops hepatic or myelo-complications after vaccination is difficult.

## 1. Introduction

Aplastic anemia (AA) is a hematological disease characterized by pancytopenia with a loss of hematopoietic stem cells, progenitor cells, and precursor cells in the bone marrow [1]. Disease severity and diagnosis are defined by the (modified) Camitta criteria [2]. A severe AA is defined by neutrophils < 0.5 G/L, platelets < 20 G/L, or reticulocytes < 20 G/L. Aplastic anemia may be congenital or caused by an abnormal immune response resulting from environmental exposures (such as chemical agents, drugs, viral infections, or auto-immune manifestation) [1]. Hepatitis-associated aplastic anemia (HAAA) is a rare but well-recognized entity in which a bone marrow failure occurs 1–3 months after an episode of acute hepatitis [3,4]. The development of various autoimmune diseases has been reported following COVID-19 infections or vaccinations. However, there is no method that can assess the relationship between vaccines and the development of autoimmune diseases. AA is an immune-mediated bone marrow failure syndrome. Nevertheless, a few cases of AA attributed to the coronavirus vaccine have been reported in the literature [5,6]. In our case, the etiology of AA was challenging as it may be related to acute hepatitis or to the coronavirus vaccine. A causative association between the COVID-19 vaccine and AA is possible, and clinicians should be aware of this rare but serious adverse event.

## 2. Case Report

A 30-year-old female patient with no past significant medical history was referred to the gastrology department for jaundice and right upper quadrant pain inspection. She had received the second dose of the coronavirus Pfizer vaccine three weeks prior to consultation. On physical examination, she was jaundiced with no signs of hepatic encephalopathy and no hepatosplenomegaly or liver tenderness or lymph nodes. Complete blood count (CBC) showed a white blood cell count of 10,500/mm^3^, 69% granulocytes, 26% lymphocytes, 2% monocytes, 3% eosinophils, 11.5 g/dL of hemoglobin, and 300,000/mm^3^ of platelets. Liver enzymes were very high (70 times the normal value) (normal < 40UI/L) without cholestasis. The prothrombin time (PT) was low (43%) [70-100%]. Abdominal ultrasonography examination was normal. Serological tests for hepatitis B virus (HBV), hepatitis C virus (HCV), hepatitis A virus (VHA), human immunodeficiency virus (HIV), cytomegalovirus (CMV), and parvovirus B19 were undetected. Autoimmune disease investigations were also negative (showing the absence of antinuclear antibodies, no hyper gammaglobulinimea, and no high immunoglobuline level). Based on these findings, a presumptive diagnosis of autoimmune-like hepatitis post-coronavirus vaccination was made. The patient was treated with intravenous corticosteroid therapy (hydrocortisone hemisuccinate at a dose of 400 mg/day), with progressive improvements in the PT and liver enzymes. However, 2 days after starting corticoids (1 week after the onset of liver disorder), the patient presented an epistaxis with gum bleeding and fever persisted despite broad-spectrum antibiotic therapy. CBC showed a white blood cell count of 500/mm^3^, 6 g/dL of hemoglobin, and 2000/mm^3^ of platelets. Bone marrow biopsy showed a poor bone marrow with no tumor infiltration or fibrosis, confirming the diagnosis of acquired AA (Figure 1). Cytogenetic analysis was normal. Based on these findings, the patient was referred for further evaluation to our hematology department. The research of paroxysmal nocturnal hemoglobinuria (PNH) clone was negative and the karyotype of peripheral blood lymphocytes was normal. Due to the severity of her aplasia (severe according to Camitta’s score) and the absence of a donor, the patient received immunosuppressive therapy (anti-lymphocyte serum (ATGAM) associated with ciclosporin) in March 2022. The evolution was marked by a progressive improvement in cytopenias (CBC after one year showing a white blood cell count of 4100/mm^3^, 63% granulocytes, 30.8% lymphocytes, 5.5% monocytes, 11.1 g/dL of hemoglobin, and 50,000/mm^3^ of platelets). The chronology of events and therapeutic schedule are detailed in Figure 2.

## 3. Discussion

AA is a life-threatening form of bone marrow failure, resulting in pancytopenia associated with bone marrow hypoplasia/aplasia. The occurrence of AA following immunizations is rare and there are few reports on vaccine-related AA in the medical literature. Two studies reported AA after receiving the recombinant hepatitis B vaccine [7,8]. The authors hypothesized that the underlying immune predisposition might have allowed the vaccines to trigger a cytotoxic T lymphocyte response that could lead to AA [9]. With regard to SARS-CoV-2 mRNA vaccinations, cases of AA following the administration of Pfizer-BioNTech’s SARS-CoV-2 mRNA vaccine have been reported [5,6]. In this case report, we describe a patient that developed presumed autoimmune hepatitis after receiving a standard COVID-19 vaccine. Shortly after the onset of hepatitis, the patient then developed an acquired AA. HAAA is a rare but well-recognized entity in which a bone marrow failure occurs 1-3 months after an episode of acute hepatitis [3]. The time interval of 1 to 3 months suggests that the initial target of the immunological response is the liver. This theory is supported by the improvement in liver function testing to immunosuppressive therapy administered for bone marrow aplasia [10]. Studies have shown that the prevalence of HAAA is 3–5% of all AA cases, and that it often affects adolescent boys and young men [3], which is not the case with our patient. The etiology of hepatitis is either idiopathic in most cases; however, in our case, due to any of the hepatitis viruses, parvovirus B19, cytomegalovirus, Epstein–Barr virus, or toxin could be induced [3].

Moreover, a study by Patel et al. [11] found that the delay between the episode of acute hepatitis and the onset of AA symptoms varies from simultaneous to 10 months (with a mild interval of 1 to 3 months). The symptoms of thrombocytopenia in our case started 1 week after the onset of jaundice and upper right quadrant pain. In all the studies, cases of AA may occur after a severe fulminant, self-limiting, or chronic hepatitis [11]. In addition, AA is seronegative most frequently [12].

The new onset of acquired marrow failure diagnosed a few weeks after SARS-CoV-2 infection in adult and pediatric patients was reported in a number of studies [13,14,15]. In patients with COVID-19 infection, the elevation of various inflammatory cytokines was found, suggesting immune dysregulation and good outcomes after immunosuppressive therapy led to the hypothesis that this coronavirus may contribute to the typical immune-mediated pathogenesis of AA [15]. 

The occurrence of AA after immunizations is rare, and there are few reports of vaccine-related AA described in the medical literature, with most cases pertaining to hepatitis B, H1N1 influenza, or varicella vaccines [5]. Our patient did not have SARS-CoV-2 infection, but she received coronavirus virus vaccination two months before the onset of pancytopenia. This vaccination delay may initiate immune-mediated marrow failure. This case does not establish a mechanistic link between SARS-CoV-2 vaccination and bone marrow failure, but it raises the possibility that the SARS-CoV-2 vaccine causes an immunologic response or, less likely, direct bone marrow toxicity, which leads to bone marrow failure. To understand the link between COVID-19 vaccination and the development of AA, it is necessary to examine the following elements: the exploration of autoantibodies against stem cells, the role of molecular mimicry between mRNA-vaccine-encoded antigens and stem cells, and the dynamics of T-cell subsets after vaccination.

The first case of AA in relation to the coronavirus vaccine was published in the *British Journal of Haematology* in 2022 [5]. It was the case of 76-year-old man, who presented with severe AA with a PNH clone one month after the second dose of the Pfizer coronavirus vaccine. Although the causality with the vaccine was difficult to demonstrate, it was the most probable trigger after eliminating all the other obvious causes of secondary AA (infection, drugs, and congenital causes). Consequently, COVID-19 vaccination may be associated with the development of AA, via several hypothetical mechanisms (molecular mimicry and aberrant over the activation of humoral and cellular immunity).

Roth et al. [16] reported that 4 out of 135 patients who were in stable hematologic remission for AA relapsed 29–42 days after receiving the second dose of the mRNA-based vaccine (Comirnaty^®^). Cecchi et al. [5] and Tabata et al. [6] reported 2 cases of 76, including a 56-year-old-man who developed AA one month following SARS-CoV-2 Pfizer-BioNTech mRNA vaccination. Sridhara et al. [1] reported the first case of a 60-year-old man with AA following the administration of the SARS-CoV-2 Moderna mRNA vaccine. These studies [1,5,6,16] concluded that the SARS-CoV-2 vaccine may be associated with the development of AA, either as a single trigger or by disclosing a latent autoimmunity. Further evaluations in large cohorts are needed to elucidate the associations between AA and SARS-CoV-2 vaccines. Clinical awareness is essential in order to evaluate patients presenting with symptoms consistent with cytopenia after receiving the SARS-CoV-2 vaccine and in order to rapidly diagnose and treat hematological disorders.

In a study by Honkaniemi et al. [12] on seven children with HAAA, liver function tests were improved when AA itself was developed. This situation was similar to our patient. In this study [12], three of the seven children had low reticulocyte counts and a low absolute neutrophil count (ANC) already at the onset of the hepatitis, which supported the hypothesis that the liver and the bone marrow were attacked simultaneously by mRNA COVID-19, though hepatic symptoms had appeared first. This may be because the exogenous agent (COVID-19 mRNA) triggers a toxic or immunological/apoptotic process that simultaneously influences the liver and the bone marrow. This case does not provide a mechanistic link between the COVID-19 vaccine and marrow failure, but it raises the possibility that the COVID-19 vaccine may mediate an immunologic response or, less likely, a direct marrow toxicity that contributes to marrow failure. Some researchers suggest that liver and marrow abnormalities in HAAA are caused by a T-cell-mediated autoimmune disease [10,17]. In our patient, the marrow aplasia responded to ATG treatment, supposing immune-related bone marrow failure and indicating a T-cell-mediated immune reactive component, at least affecting the bone marrow. 

The treatment of AA depends on the age of the patient, as well as the presence or absence of an HLA identical intrafamily donor. For patients younger than 45 years old with an HLA-matched sibling, allogeneic bone marrow transplantation is considered the gold standard. Otherwise, patients are treated with anti-thymocyte globulin associated with cyclosporine. The prognosis of AA has improved in recent years due to the increasing availability of allogeneic bone marrow transplantation, immunosuppressive therapy, and supportive care [18]. The mechanism for these rare conditions and the relationship with vaccination are not clear and require further study.

## 4. Conclusions

A causative association between coronavirus vaccination and immune manifestations is well recognized. However, in our case, although the episode of acute hepatitis was related to the earlier administration of the COVID-19 vaccine, it is unknown whether the preceding acute hepatitis may have precipitated AA or whether the COVID-19 vaccine was the inciting agent in both rare events. In any case, several reports of AA complication caused by the COVID-19 vaccine should be taken into consideration when weighing the risks and benefits of COVID-19 vaccination.

## Figures and Tables

**Figure 1 vaccines-11-01228-f001:**
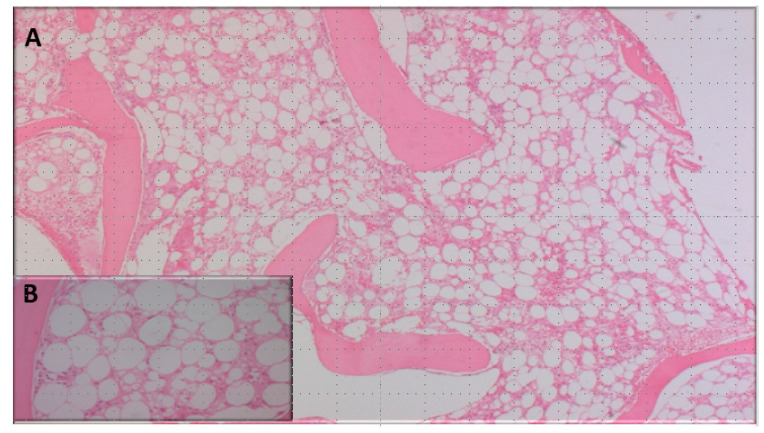
Microphotography showing markedly hypocellular bone marrow devoid of hematopoietic cells replaced with fat cells and only scattered lymphocytes, plasma cells, and mast cells without fibrosis, hemosiderosis, or tumoral cells (HE: hematoxylin eosin; (**A**) *10, (**B**) *40).

**Figure 2 vaccines-11-01228-f002:**
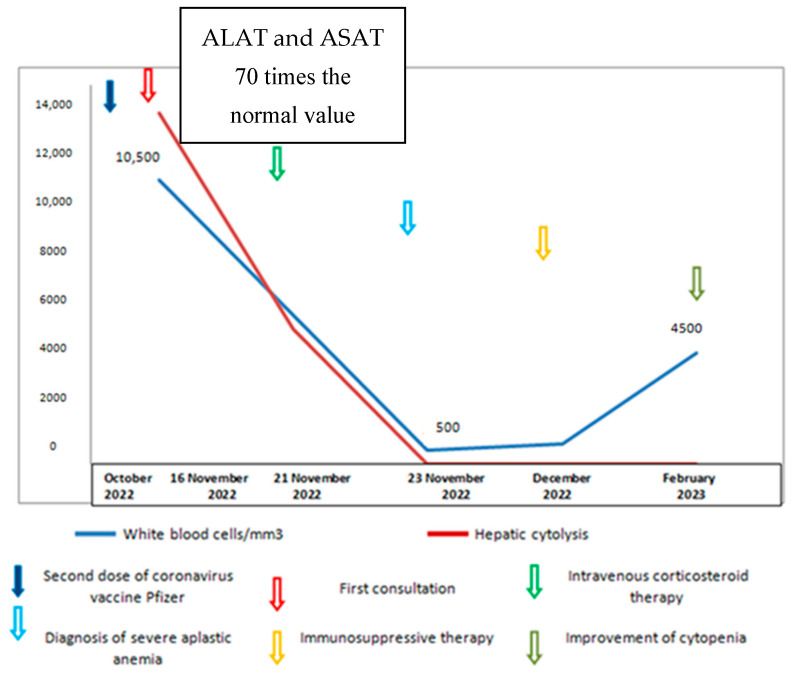
The chronology of events and therapeutic schedule.

## Data Availability

The dataset of this study is available from the corresponding author upon reasonable request.

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
