# Peer review of "A Case of Acquired Aplastic Anemia after Severe Hepatitis- Probably Induced by the Pfizer/BioNTech Vaccine: A Case Report and Review of Literature"

_vaccines, 2023, doi:10.3390/vaccines11071228_

Round 1

Reviewer 1 Report

The article demonstrated a rare case of aplastic anemia after acute severe hepatitis which occurred three weeks after the mRNA SARS-CoV-2 vaccination. Hematologists occasionally see transient pancytopenia after vaccination. The case presented might be vaccine-related myelosuppression (References; https://doi.org/10.1002/jha2.443, https://doi.org/10.1016/j.jaut.2021.102782). As the authors indicated, another etiology, in this case, is hepatitis-associated aplastic anemia. It is very hard to distinguish the cause of aplastic anemia between vaccine- and hepatitis-related AA.

Would you add data on aspartate and alanine transaminases in Figure 2 to better understand the degree of ‘hepatic cytolysis’?

Showing the authors’ advice to the patient about the booster vaccination would be appreciated.

There is a slight difference in the descriptions in the conclusion between the abstract and the main text. Authors recommend ‘close monitoring’ after vaccination; however, predicting who develops hepatic or myelo-complications after vaccination is tough. 

English is fine; however, minor editing issues exist, like 'hemoglobin 6/dL' would be '6 g/dL'

Author Response

Dear Reviewer,

  • data on aspartate and alanine transaminases are added in Figure 2
  • Introduction is improved.
  • Conclusion of the abstract is modified.
  • Language editing: 'hemoglobin 6/dL' is corrected to be '6 g/dL'
  • Best Regards

Reviewer 2 Report

This reviewer is a clinician and a clinical research investigator. And as the reviewer I wanted to first compliment this team on their "state of the art" thorough diagnostic evaluation and successful treatment of their patient. Their efforts saved this patient's life! 

Now the case report adds to existing literature related to the very rare potential serious adverse events of hepatitis and aplastic anemia. This is not novel as these adverse effects have been noted for other vaccines and specifically for the COVID-19 mRNA/ protein vaccine. The interesting but unanswered question is what is the mechanism and is the bone marrow response secondary to the hepatitis or due to the vaccine directly. The case report suggests this association but does not provide any speculative mechanism related to the vaccine effects. The other issue is whether these effects are related to the mRNA delivery of the spike protein or an immune immune reaction per se. 

It would improve the quality of the report for the authors to review the current literature and propose potential mechanisms for these events related to this COVID-19 vaccine. Additionally countries have adverse events reporting registries (USA-MedWatch FAER, Europe Eudra Vigilance) and companies - Pfizer/ Moderna etc. It would have been  interesting if reports from these registries are available to provide a larger context to that from case reports in the literature 

I have uploaded an edited version- my suggested rewrites are embedded as comments in the PDF.

There are a number of English non-standard phrasing/ terms and syntax that can be readily corrected 

Author Response

Dear reviewer

thank you very much.

the comments are considered

Best regards

Round 2

Reviewer 2 Report

Thank you for the revisions and edits based on my comments